# Cholangiocarcinoma: The Current Status of Surgical Options including Liver Transplantation

**DOI:** 10.3390/cancers16111946

**Published:** 2024-05-21

**Authors:** Abdullah Esmail, Mohamed Badheeb, Batool Alnahar, Bushray Almiqlash, Yara Sakr, Bayan Khasawneh, Ebtesam Al-Najjar, Hadeel Al-Rawi, Ala Abudayyeh, Yaser Rayyan, Maen Abdelrahim

**Affiliations:** 1Section of GI Oncology, Department of Medicine, Houston Methodist Cancer Center, Houston, TX 77030, USA; 2Department of Internal Medicine, Yale New Haven Health, Bridgeport Hospital, Bridgeport, CT 06605, USA; 3College of Medicine, Almaarefa University, Riyadh 13713, Saudi Arabia; 4Zuckerman College of Public Health, Arizona State University, Tempe, AZ 85287, USA; 5Department of GI Medical Oncology, Division of Cancer Medicine, The University of Texas MD Anderson Cancer Center, Houston, TX 77030, USA; 6Faculty of Medicine, The University of Jordan, Amman 11942, Jordan; 7Division of Internal Medicine, The University of Texas MD Anderson Cancer Center, Houston, TX 77030, USA; 8Department of Gastroenterology & Hepatology, Faculty of Medicine, The University of Jordan, Amman 11942, Jordan

**Keywords:** cholangiocarcinoma, intrahepatic cholangiocarcinoma (iCCA), extrahepatic cholangiocarcinoma (eCCA), liver transplantation, OLT, liver resection and PVE

## Abstract

**Simple Summary:**

Cholangiocarcinoma (CCA) poses a substantial threat as it ranks as the second most prevalent primary liver tumor. This review reports comprehensively describe the surgical strategies available for treating CCA, including the preoperative measures and interventions, alongside the current options about liver resection and OLT. CCAs are aggressive primary liver tumors with an average 5-year survival of 5% in nodal-positive disease. The primary surgical approach should aim for complete R0 resection along with lymphadenectomy for right staging. However, even in patients who have undergone tumor re-section, the five-year survival rate remains low at around 25%. On the other hand, the combined neoadjuvant OLT showed an increase in the five-year survival rate when compared to surgical resection.

**Abstract:**

Cholangiocarcinoma (CCA) poses a substantial threat as it ranks as the second most prevalent primary liver tumor. The documented annual rise in intrahepatic CCA (iCCA) incidence in the United States is concerning, indicating its growing impact. Moreover, the five-year survival rate after tumor resection is only 25%, given that tumor recurrence is the leading cause of death in 53–79% of patients. Pre-operative assessments for iCCA focus on pinpointing tumor location, biliary tract involvement, vascular encasements, and metastasis detection. Numerous studies have revealed that portal vein embolization (PVE) is linked to enhanced survival rates, improved liver synthetic functions, and decreased overall mortality. The challenge in achieving clear resection margins contributes to the notable recurrence rate of iCCA, affecting approximately two-thirds of cases within one year, and results in a median survival of less than 12 months for recurrent cases. Nearly 50% of patients initially considered eligible for surgical resection in iCCA cases are ultimately deemed ineligible during surgical exploration. Therefore, staging laparoscopy has been proposed to reduce unnecessary laparotomy. Eligibility for orthotopic liver transplantation (OLT) requires certain criteria to be granted. OLT offers survival advantages for early-detected unresectable iCCA; it can be combined with other treatments, such as radiofrequency ablation and transarterial chemoembolization, in specific cases. We aim to comprehensively describe the surgical strategies available for treating CCA, including the preoperative measures and interventions, alongside the current options regarding liver resection and OLT.

## 1. Introduction

Cholangiocarcinoma (CCA) represents a diverse range of malignancies that can develop anywhere along the biliary system, from the minor canals of Hering to the primary bile duct. These cancers are anatomically classified as intrahepatic (iCCA), perihilar (pCCA), and distal CCA (dCCA). Despite sharing similarities, they also manifest significant distinctions between different tumors and even within individual tumors, influencing their origins and clinical outcomes [1]. CCA stands as the second most frequent primary liver tumor, representing roughly 10–15% of all hepatobiliary malignancies [2]. Worldwide, there is a rising trend in both the incidence and mortality of CCA, with the highest rates observed in Asian populations in comparison to Western regions. Recent mortality patterns indicate a disparity between different subtypes of CCA, showing an increase in mortality rates for iCCA as opposed to extrahepatic cholangiocarcinoma (eCCA) (Figure 1) [3]. In the United States, there was a documented yearly rise of 5.9% in the incidence of iCCA from 2003 to 2009 [4]. Despite undergoing tumor resection, the five-year survival rate remains low at 25%, primarily due to tumor recurrence, which accounts for the leading cause of death in 53–79% of patients. Recurrence typically happens within the first two years post-tumor resection and is commonly localized in 83% of cases [5,6,7,8,9,10,11].

Evaluating the eligibility of surgical treatment requires the collaboration of a specialized team comprising oncologists, surgeons, gastroenterologists, radiologists, and pathologists. This approach is crucial for determining the most appropriate treatment plan. Assessing a patient’s performance status and risk factors plays a vital role, as it helps identify individuals with sufficient liver function capacity and reduces the risk of peri-operative mortality [12]. The effectiveness of surgical resection for CCA relies on several factors, including tumor differentiation, tumor size, the number of tumors, surgical margin, and the extent of lymph node metastasis [9]. Novel multifactorial models have been introduced to refine the criteria for patient selection in surgical resection. These models are primarily centered on utilizing preoperative laboratory parameters that are easily obtainable, with the goal of identifying those individuals for whom surgical resection would yield the most significant clinical benefit [13]. Moreover, in most medical institutions globally, orthotopic liver transplantation (OLT) has conventionally been avoided as a treatment option for iCCA due to markedly unfavorable initial outcomes, characterized by a 2-year survival rate of approximately 30%. These suboptimal results can be attributed to the absence of uniform patient selection criteria and the limited use of neoadjuvant therapies. A significant shift in this landscape emerged when precise patient selection criteria were demonstrated, particularly in enhancing survival outcomes in OLT for early-stage liver cancers [14,15,16,17,18,19,20,21,22]. Therefore, understanding the evolving landscape of surgical treatment options becomes increasingly critical. This review article discusses the available surgical strategies for managing CCA, aiming to shed light on the advancements, limitations, and emerging trends in the field. In addition to providing a thorough analysis of surgical interventions, this review seeks to contribute to the knowledge base essential for clinicians and researchers in their efforts to improve patient outcomes in the face of this formidable disease.

## 2. Diagnostic Evaluation

Pre-operative evaluations of iCCA are crucial and usually institutionally driven, with the primary goal of detecting the precise tumor location, biliary tract involvement, vascular encasements, and nodal or distant metastasis [23]. Typically, carbohydrate antigen (CA)19-9, cross-sectional imaging-contrast-enhanced computed tomography (CT), and/or magnetic resonance imaging (MRI) are obtained prior to any surgical intervention [24]. The primary tumor marker employed for CCA is CA19-9, exhibiting a sensitivity of approximately 90% and a specificity that can reach up to 98% when considering a cutoff of 100 U/mL [25]. Interestingly, occult metastatic CCA has been linked to CA19-9 levels > 1000 U/mL [26]. A fresh study revealed that patients with locally advanced (*p* < 0.0001) or metastatic (*p* < 0.0001) iCCA had higher serum levels of carcinoembryonic antigen (CEA) (cutoff value: 5 U/mL) and CA19-9 (cutoff value: 37 U/mL) than patients with earlier stage liver-confined disease [27]. Furthermore, there have been documented instances suggesting the potential prognostic value of CA19-9 and CEA in iCCA [28,29].

When evaluating patients with obstructive jaundice or abdominal pain, ultrasound is frequently the first imaging modality employed. It can be used to rule out choledocholithiasis and diagnose CCA. With 64.1% sensitivity, 97.4% specificity, and 73.6% accuracy, contrast-enhanced ultrasonography can assist in differentiating hepatocellular carcinoma (HCC) from iCCA [30]. However, a multiphasic, contrast-enhanced, multidetector-row CT scan or a contrast-enhanced MRI/magnetic resonance cholangiopancreatography (MRCP) scan is usually the first imaging test performed in patients with suspected iCCA and no prior liver-specific imaging studies. This approach is particularly relevant, for instance, when elevated CA19-9 levels are detected in the patient [31]. Moreover, MRCP demonstrates a sensitivity of 87% and a specificity of 85% in distinguishing between benign and malignant origins of hilar obstruction [32]. Endoscopic retrograde cholangiography (ERCP) serves a dual diagnostic and therapeutic function in pCCA and dCCA. It enables the identification of malignant strictures and the acquisition of biliary brushings for cytology and fluorescence in situ hybridization (FISH) analysis [33]. In Table 1, we summarize various imaging modalities used for the preoperative evaluation of CCA.

The diagnosis of CCA can be established through biliary cytology or by confirming the presence of adenocarcinoma in a biopsy sample. However, a meta-analysis revealed that biliary cytology exhibits limited sensitivity, specifically 43%, in pCCA detection. For the detection of CCA, FISH analysis demonstrates comparable specificity and superior sensitivity compared to cytology (65% versus 19% in one series) [33,34]. While conventional diagnostic methods for iCCA do not typically involve a liquid biopsy, the emergence of innovative techniques, such as the detection of circulating tumor deoxyribonucleic acid (ctDNA), has offered promising approaches to address these diagnostic challenges [35]. In the context of CCAs, the extensive sequencing of peripheral blood samples, encompassing both whole-exome and targeted gene analysis, has revealed that TP53 and Kirsten rat sarcoma viral oncogene homolog (KRAS) are among the genes with the highest frequency of mutations. Additionally, epithelial growth factor receptor (EGFR) alterations have been documented [36]. Likewise, genetic alterations in biliary tract cancers can be directed toward targets involving IDH1, FGFR2, and v-raf murine sarcoma viral oncogene homolog B1 (BRAF) [37]. Figure 2 shows the prevalence of various genetic alterations by anatomic classification in CCA [38].
cancers-16-01946-t001_Table 1Table 1Different imaging modalities that can be used for the preoperative evaluation of CCA. Endoscopic retrograde cholangiopancreatography (ERCP), percutaneous transhepatic cholangiography (PTC).TechniqueSensitivitySpecificityAdvantagesLimitationsUltrasound87–96% [39]--Affordable and readily available.-Obtaining tissue through endoscopic ultrasound with high sensitivity [40].-Infiltrative pCCA and eCCA are often difficult to detect.-Operator-dependent.-The distal bile duct can be obscured by GI gas or fat.CT scan61% [41]88% [41]-Identification of hepatic masses.-Detection of bile or pancreatic duct and vascular encasement.-Evaluating nodal or distant metastases.-Ionizing radiation-Harmful to overall patient health.-Limited capacity to assess the tumor’s progression through the biliary tract.-Lower sensitivity in detecting nodal involvement preoperatively than MRI [42].MRI90–97% [43]60–81% [43]-Highly accurate, non-invasive.-No ionizing radiation.-Provides an accurate depiction of bile ducts and surrounding structures [44,45].-Long scanning time.-Severe claustrophobia, clips, pacemakers, and other metallic implants are contra-indications to MRI [46].Direct Cholangiography (ERCP/PTC)46–73% [47,48]100% (with tissue sampling) [47,48]-Enables direct visibility of the biliary system.-Help distinguish between ambiguous bile duct strictures.-Permits tissue sampling for histopathological diagnosis.-Offers therapeutic drainage [49].-Invasive.-Operator-dependent.-Intestinal or biliary leakage and perforation, cholangitis, pancreatitis, and bleeding [49].Direct single operator cholangioscopy combined with (DSOC)85–86% [47,48,50]100% (with tissue sampling) [47,48,50]-Enables the direct visualization of the biliary system.-Helps distinguish between ambiguous bile duct strictures.-Permits direct tissue sampling for histopathological diagnosis.-Improved sensitivity over ERCP with biopsy or brushing.-Invasive, requires ERCP.-Operator-dependent.-Intestinal or biliary leakage and perforation, cholangitis, pancreatitis, and bleeding.Linear endoscopic ultrasound with fine needle aspiration(EUS-FNA)43–89% (Better for dCCA than pCCA) [51,52,53].79–100% [53]-Allows for tissue sampling.-Maybe a safer alternative to ERCP.-Recommended for cases with inconclusive brushing results on ERCP.-100% positive predictive value.-Invasive.-Low negative predictive value (29–67%).-Bleeding, pancreatitis, perforation and infection.-Peritoneal metastasis with FNA [54].Intraductal sonography(IDUS)89–91% [55].80–92% [55]-More accurate than ERCP with biopsy to distinguish between benign and malignant strictures.-Better than ERCP to assess longitudinal tumor spread.-High accuracy for invasion of the portal vein and right hepatic artery-Invasive.-Image interpretation is challenging.-Limited in lymph node assessment.-No tissue sampling.Confocal laser endomicroscopy(CLE)88% [56]79% [56]-More accurate than ERCP with biopsy in indeterminate biliary strictures-Imagining interpretation is challenging, but may improve with AI/machine learning [56].-Decreased accuracy with biliary stent placement [57].

## 3. Surgical Management

CCA is well-recognized for its poor prognosis, with an average 5-year survival of 5% in nodal-positive cases. Individual patients and tumor-related factors heavily influence the approach to managing CCA. These considerations include age, performance status, the overall liver health of the patient, and the extent of the disease, which can range from solitary and locally resectable lesions to locally unresectable or metastatic conditions. Ideally, a multidisciplinary team collaborates to assess patients for various therapeutic options, including surgical resection, perioperative chemotherapy, liver-targeted interventions, transplantation, and systemic therapies such as cytotoxic treatment, targeted therapy, and immunotherapy [58]. The surgical approaches primarily include hepatic resection or OLT.

### 3.1. Preoperative Intervention

#### 3.1.1. Staging Laparoscopy

Preoperative imaging may not always show evidence of invasive or locally advanced illness, including peritoneal seeding. In patients with hidden metastatic illness, the use of staging laparoscopy at the beginning of a surgical treatment helps to avoid needless laparotomies. A retrospective analysis revealed that metastatic disease was identified in 29% of patients with hepatobiliary malignancies, including iCCA, during staging laparoscopy. Consequently, one out of every five patients did not require laparotomy, mitigating the associated increases in hospitalization duration and morbidity [59]. In a different study, 36% of patients subjected to staging laparoscopy for CCA or gallbladder cancer exhibited advanced or unresectable disease [60]. However, certain physicians opt for a selective approach to laparoscopy, primarily for patients at elevated risk, such as those with high CA 19-9 levels or indeterminate radiographic findings, as advised by an expert consensus statement issued by the American Hepato-Pancreato-Biliary Association.

#### 3.1.2. Preoperative Biliary Decompression

Consensus regarding pre-operative biliary decompression is lacking, including the preferred timing or method [61,62,63,64]. Nevertheless, it is commonly performed in patients with pCCA with high bilirubin levels (more than 10 mg/dL); a [63,65,66] study showed that PBD was useful in decreasing bilirubin levels and reversing cholestatic liver dysfunction. The main approaches used in PBD are either endoscopic or percutaneous. The endoscopic approach is mainly carried out via the Endoscopic Retrograde Cholangiopancreatography (ERCP) method and, to a lesser extent, Endoscopic Ultrasound (EUS)-assisted methods [62]. Such an intervention is thought to limit the postoperative morbidity and mortality that is linked to unresolved cholestasis, including hepatic and biliary cirrhosis [67]. On the other hand, the percutaneous approach consists of Percutaneous Transhepatic Biliary Drainage (PTBD). ERCP is less invasive, but it has a higher risk profile including ascending cholangitis, duodenal perforation, and pancreatitis. PTBD, on the other hand, carries the risk of complications such as bleeding, tumor seeding, portal vein thrombosis, and feelings of discomfort [63,65,66]. PTBD has shown superiority compared to ERCP in preoperative settings in pCCA patients. While there is increasing interest among gastroenterologists towards EUS-guided biliary decompression, the most common adverse events were bleeding, peritonitis, cholangitis, and pneumoperitoneum. However, it has limited application due to an insufficient number of endosonography professionals [63,65,66]. PBD is thought to limit the postoperative morbidity and mortality that is linked to unresolved cholestasis, including hepatic and biliary cirrhosis. On the other hand, there are concerns regarding the real benefits of preoperative biliary drainage and whether it increases the risk of severe infectious complications, such as cholangitis, in pCCA patients. Ameta-analysis by Liu et al. showed no difference in hospital stay and the overall increase in post-operative infection rate [68]. However, we believe that biliary drainage provides symptomatic relief (e.g., pruritus) in patients awaiting surgery, in addition to the normalization of hepatic function, which may be needed for neoadjuvant chemotherapy recipients. Moreover, biliary drainage may be necessary for CCA patients who present with cholangitis preoperatively [69].

#### 3.1.3. Preoperative Portal Vein Embolization (PVE)

Several studies have shown improved survival, liver synthetic functions, and overall mortality with PVE [70,71,72,73,74]. The rationale behind PVE is to permit compensatory hypertrophy in the unaffected hepatic lobes in patients who are anticipated to have an insufficient hepatic remnant (Figure 3). PVE prolongs the period before resection by at least 3–6 weeks to enable enough development of the predicted remaining hepatic tissue [75].

### 3.2. Hepatic Resection

The surgical approach in CCA varies based on the tumor’s location. Curative-intent treatment for CCA primarily relies on surgical resection, which offers the potential for a complete cure [76]. In the case of iCCA, it often involves the major hepatic veins or the inferior vena cava. The standard procedure typically involves a negative-margin resection of affected liver segments or lobes. Concomitant portal lymph node resection is also advised to achieve the best survival outcomes. Additionally, complete vascular resection, ex situ surgery, and reconstruction might be required for the vessel resection of affected liver segments or lobes. For dCCA, pancreatoduodenectomy, also known as the Whipple procedure, is commonly performed [77,78,79,80,81,82]. pCCA resection presents more complexity due to its central location within the liver. The pCCA frequently involves the portal vein and the hepatic artery. An analysis found that main portal vein or bilateral portal vein involvement was associated with lower survival rates as compared to unilateral portal vein involvement. Portal vein surgical removal and reconstruction is considered a safe and beneficial procedure. It is a standard option whether through direct connection or jump grafts. On the other hand, hepatic artery resection is a challenging procedure due to its higher morbidity [77,78,79,80]. In order to avoid the necessity for artery reconstructions, collateral systems or arterio-portal shunts are available. The resecting of pCCA presents greater complexity due to its central location within the liver. Consequently, surgical intervention includes both intrahepatic and extrahepatic bile duct resection coupled with major hepatic and caudate lobe resection. Porta hepatis lymph nodes must also be resected to ensure thorough tumor resection [77,78,79,80,81,82,83]. Obtaining tumor-free margins is achievable in only 20% and about 50% of patients with pCCA and dCCA, respectively [13]. The key factors influencing post-resection survival include the achievement of tumor-negative margins, the absence of vascular invasion and lymph node metastasis, and the preservation of adequate functional liver tissue [84]. The reported 5-year survival rates after resection fall within the range of 22–44% for iCCA, 11–41% for pCCA, and 27–37% for dCCA [83].

#### 3.2.1. General Considerations in Hepatic Resection

aCriteria of resectability

The evaluation of surgical eligibility for iCCA encompasses a comprehensive assessment spanning various anatomic, physiological, and biologic domains. Anatomic resectability entails the capacity to fully excise the affected liver portion while ensuring the presence of a sufficient future liver remnant (FLR). Physiologic resectability pertains to the patient’s performance status, comorbidities, and general capacity to endure a significant surgical procedure with an acceptable risk of experiencing complications [85]. Different criteria have been implemented to provide an appropriate patient selection; for example, Schulick endorsed surgical resection in medically fit patients with appropriate preoperative evaluation for their medical history, hepatic functions, metastasis involving the peritoneal cavity, chest, and other sites if indicated. In addition, the tumor must be evaluated radiologically for vascular flow, parenchymal alterations, and biliary involvement [86]. Other traditional criteria include the lack of involvement of the main portal vein or hepatic artery, retropancreatic and paraceliac nodal metastases or distant metastases, and extrahepatic adjacent organ invasion [87,88,89,90]. These are conventional criteria, and many skilled facilities perform resection on cases previously considered unresectable.

bFuture liver remnant (FLR)

A critical preoperative factor in managing iCCA is assessing the patient’s functional capacity and the adequacy of their FLR. Hepatectomy, as a complex surgical procedure, places substantial physiological demands on the patient’s body. Individuals with multiple underlying health issues may lack the necessary physiological reserves for effective recovery or the management of potential complications. Upon establishing a patient’s eligibility for surgery, ensuring that the FLR is of adequate size becomes imperative to mitigate the risk of postoperative hepatic insufficiency [91]. To evaluate the FLR, liver volume can be quantified using CT or MRI. Occasionally, there can be a disparity between FLR volume and postoperative hepatic function. Technetium-99m mebrofenin hepatobiliary scintigraphy is a quantitative diagnostic tool that may provide valuable insights into FLR function [92]. Furthermore, a quantitative assessment of liver function can be achieved through an indocyanine green (ICG) clearance test to predict hepatic functional reserve. A retention rate of 15 min has been linked to the risk of post-hepatectomy liver failure. Associating Liver Partition and Portal vein ligation for Staged hepatectomy (ALPPS) has been found to be superior to traditional hepatectomy because it induces a strong ability for liver regeneration and hypertrophy, which is crucial in patients with insufficient FLR. As compared to ALPPS, partial ALPPS is a variation that carries a lower risk of morbidity and mortality [93].

cLymphadenectomy

Lymph node metastases carry significant prognostic implications for the survival of iCCA patients [7,94,95]. In patients who have undergone surgical resection for iCCA, lymph node metastases were detected in 45–65% of cases and represent one of the most significant prognostic factors following the surgical procedure [96]. However, surgical resection for patients with lymph node metastasis did not cause an improvement in median survival compared to patients who were treated with chemotherapy alone [97]. On the other hand, a retrospective study involving patients who underwent surgical resection for clinically node-negative iCCA but were found to be node-positive at pathology demonstrated that performing a thorough lymphadenectomy (specifically, retrieving at least six lymph nodes) substantially enhances survival and reduces the risk of tumor recurrence [98]. Furthermore, the guidelines provided by the National Comprehensive Cancer Network (NCCN) advocate portal lymphadenectomy during resection to facilitate precise staging. Specifically, the evaluation should encompass a minimum of six lymph nodes, and the scope of lymphadenectomy should include the dissection of lymph nodes along the common hepatic artery and within the hepatoduodenal ligament [99,100]. The decision on the scope of lymphadenectomy in iCCA surgery should be guided by a comprehensive knowledge of the liver’s lymphatic drainage patterns concerning right-sided versus left-sided iCCA (Figure 4) [101]. Some experts recommend selective portal dissection for central iCCA, a procedure that may require additional extra-hepatica bile duct resections along with lymph nodes. Almost half of the patients who are identified as having resectable iCCA are found to be ineligible for resection when explored surgically; thus, staging laparoscopy has been suggested to limit unnecessary laparotomy, as mentioned earlier [102,103].

#### 3.2.2. Intrahepatic CCA

Obtaining negative resection margins is the goal of surgical resection in iCCA [104]. Although the supporting evidence for such a practice is not universal—given that many studies supported the importance of negative margins (R0) as the best predictor of mortality and recurrence predictors while others did not—the current evidence endorses targeting the goal of obtaining negative margins following resection [105,106,107,108]. The silent nature and late presentation of iCCA render this process arduous and require surgical resections that extend to involve adjacent vascular and biliary structures [109]. The complexity and difficulty of such interventions may explain the high recurrence rate of iCCA in up to two-thirds of cases within one year and an approximate median survival of ≤12 months for recurrent cases [110,111]. The treatment of iCCA used to be palliative; however, surgical resection was first introduced by the European Association for the Study of the Liver (EASL) recommendations in 2014 [112]. Several studies were published later, supporting the utility of surgical resection, either minor hepatectomies or segmentectomies, in improving survival. For instance, a multicentric study in Germany found a median survival of 98%, 78%, and 57% in 1-, 3-, and 5-year periods, respectively [113]. Other studies [111,114,115,116] endorsed these findings, highlighting the role of hepatic resection and repeated resections in iCCA management.

Nevertheless, iCCA recurrence following repeated resections is not uncommon; in fact, such cases are identified in proximity to surgical or hilar structures, limiting the feasibility of repeating surgical resections [10,110]. Around 25% of patients who had a curative-intent hepatectomy for iCCA experienced very early recurrence (within 6 months), and this was linked to a significantly poor prognosis. The 5-year overall survival for those with very early recurrence was only 8.9%, compared to 49.8% for those without it [114]. Moreover, to secure an R0 resection, it may be necessary to perform complex vascular resection. This procedure is attainable and safe for certain patients at high-volume, experienced medical centers. In a multicenter study involving iCCA resection patients, those who underwent major vascular resections exhibited similar perioperative outcomes, including complications of any type and major complications, as well as oncologic outcomes such as recurrence-free survival and overall survival, when compared to patients who did not require vascular resections. Simultaneous major vascular resection should be considered for well-suited iCCA patients undergoing hepatectomy, but it is advisable to conduct these surgeries in high-volume medical facilities [117].

#### 3.2.3. Peri-Hilar CCA

For patients with pCCA, achieving complete R0 surgical resection offers the most promising opportunity for long-term survival. However, less than 40% of patients initially present with resectable disease, and during surgical exploration, almost half of these individuals are found to have unresectable disease [24]. The typical surgical approach for pCCA includes a right extended hepatectomy or left hemi-hepatectomy, accompanied by the resection of the bile duct and lymphadenectomy in the porta hepatis region, as well as bilioenteric reconstruction. The inclusion of caudate resection enhances the chances of achieving R0 resection and improved survival without introducing significant additional morbidity [118]. Bile duct resection is insufficient in pCCA, as hepatic duct confluence is involved early [119]. Thus, concurrent hepatic resection has been associated with better outcomes [120,121,122].

Generally, the surgical approach for pCCA is primarily based on the Bismuth–Corlette classification: (a) Type I is limited to the common hepatic duct, below the confluence level of the right and left hepatic ducts. (b) Type II involves the confluence of the right and left hepatic ducts. (c) Type IIIa extends to involve the origin of the right hepatic duct. (d) Type IIIb extends to involve the origin of the left hepatic duct. (e) Type IV extends to and involves the origins of both right and left hepatic ducts. For instance, extra-hepatic duct and gallbladder en bloc resection is performed for type I and II with 5–10 mm margins, in addition to regional lymphadenectomy with Roux-en-Y hepaticojejunostomy reconstruction compared to hilar en bloc resection, which involves hepatic lobectomy with multiple hepatic segments and portal vein resections, for type III and IV, respectively [87,123,124,125,126]. En-bloc resection, which includes either hepatectomy or pancreaticoduodenectomy, offers an increased probability of achieving an R0 resection and enhanced survival compared to isolated bile duct resection [121]. On the other hand, when tumors exhibit significant bile duct involvement, a combination of pancreaticoduodenectomy and hepatectomy may be necessary to attain negative margins. However, it is important to note that the mortality rate associated with this combined procedure can be substantially elevated, to approximately around 10%. Patients who underwent combined pancreaticoduodenectomy and hepatectomy and achieved an R0 resection exhibited 5-year survival rates within a range of 18% to 68%. Conversely, patients with positive margins did not survive beyond 5 years [127]. Moreover, a significant difference in the 5-year survival rate is primarily evident when comparing R0 resections (with up to 60% survival) to individuals who have not undergone surgery. However, this discrepancy is contingent upon the lymph node status. In cases with negative lymph nodes, the 5-year survival can reach up to 55%, while in cases with positive lymph nodes, it tends to be approximately 20% [128]. Due to the increased demand for high levels of precision and accuracy in the surgical treatment of pCCA, Robotic Surgery (RS) is currently being studied as a possible treatment option for pCCA. A recent study showed promising outcomes regarding RS, with the main drawback being its prolonged operation times. However, in the meantime, surgical resection remains the only curative treatment option despite possible risks for morbidity and mortality [78].

### 3.3. Orthotopic Liver Transplantation (OLT)

Unresectability is not uncommon in CCA; in fact, the majority of CCA tumors are diagnosed in advanced stages, yielding a low resectability rate. Patients who are ineligible for surgical resection due to factors such as tumor size, location, or liver disease can have OLT as a potential alternative treatment option [129,130]. Although OLT used to be contraindicated in hilar CCA, the introduction of neoadjuvant radio-chemotherapy, which was implemented and supplemented with OLT by the Mayo Clinic, shed light on the role of OLT in CCA management [18,131,132,133,134,135,136,137]. Common reasons for failing to attain R0 resection include difficulties in identifying the hepatic duct and removing implicated adjacent tissues, such as the caudate lobe. Additionally, the benefit of being able to keep a greater hepatic volume as well as removing the tumor with the largest possible margin, while removing the pro-carcinogenic micro-environment, makes OLT a preferable alternative to resection [138,139]. In Table 2, we summarized prospective trials of OLT in CCA. Additionally, we have to take into consideration the optimization of OLT outcomes in light of the limited availability of liver grafts. Currently, some groups have proposed performing OLT in patients with resectable hilar cholangiocarcinoma as a therapeutic option.

#### 3.3.1. OLT Eligibility

Given such findings, the United Network for Organ Sharing (UNOS) decided to provide an exemption for CCA patients who received neoadjuvant therapy from the Model of End-Stage Liver Disease (MELD) score [140]. To be eligible, patients must have an unresectable tumor and/or a tumor with a diameter of ≥3 cm, without metastases, with pathological tissue or cytology, confirmation of malignancy, and a CA-19-9 > 100 U/mL. In addition, patients should have no cholangitis or aneuploidy and receive neoadjuvant chemoradiation with a thorough workup with imaging and staging laparotomy prior to OLT [131,132,133,134,135,139]. The impact of neoadjuvant therapy on tumor downstaging and its potential to allow for R0 resection for initially unresectable tumors may have promising outcomes in transplant literature. However, more studies are needed in this regard as there is currently no indication to delay tumor resection for neoadjuvant therapy; see Table 3 and Table 4 [138,141].

#### 3.3.2. Intrahepatic CCA

Historically, OLT was infrequently considered for iCCA due to its aggressive characteristics, with initial data showing a modest 5-year survival rate ranging from 10% to 18%. Currently, HCC predominates as an OLT indication, yet there is a renewed interest in evaluating the suitability of iCCA for OLT [16,17,18,20,21,22,137,142,143]. Recent findings indicate that OLT could be a viable choice for individuals with small iCCA tumors. A multicenter retrospective study revealed a 5-year actuarial survival rate of 65% after OLT in patients with cirrhosis and small, incidental iCCA tumors measuring less than 2 cm [144]. Moreover, Sapisochin G. et al. [145] investigated outcomes in cirrhotic patients who, following OLT for HCC, were discovered to have a combination of HCC and CCA or iCCA upon pathological examination. The study revealed comparable 5-year overall survival rates of 73% for patients with a single iCCA tumor measuring 2 cm or less compared to those with HCC. However, OLT in individuals with large iCCA tumors carries a significant risk of tumor recurrence (Figure 4 and Figure 5).

Lunsford et al. [146] documented a case series involving six patients diagnosed with iCCA who underwent neoadjuvant chemotherapy utilizing gemcitabine before OLT. The reported outcomes indicate a one-year overall survival rate of 100%, with survival rates of 83.3% at both the three- and five-year marks. Recurrence of the disease was observed in three patients at a median time of 7.6 months post-OLT. McMillan et al. [147] indicated that in patients with locally advanced and unresectable iCCA who received neoadjuvant chemotherapy, achieving six months of disease stability without extrahepatic involvement before undergoing OLT resulted in notable survival rates: 100% at one year, 71% at three years, and 57% at five years. Afterward, Abdelrahim et al. [16] conducted a study involving 10 patients diagnosed with iCCA or hilar CCA who underwent OLT. Prior to OLT, these patients received a neoadjuvant therapy regimen combining gemcitabine and cisplatin for 181 days. The results revealed that nine out of ten patients (90%) did not experience recurrence or metastasis, while only one patient (10%) had confirmed metastasis. The overall survival rates were 100% for both one and two years and 75% for years three to five. Despite these promising findings, OLT for iCCA is not currently endorsed as a standard treatment. Nevertheless, ongoing clinical trials seek to establish the utility of OLT for this specific patient group and to identify optimal candidates for this approach [15,22,58,148,149].
cancers-16-01946-t003_Table 3Table 3Summary of the studies investigating intrahepatic cholangiocarcinoma in the liver transplant setting. Inj: injection; KM: Kaplan–Meier; LT: liver transplant; PSC: primary sclerosing cholangitis; RFA: radiofrequency ablation; TACE: transarterial chemoembolization; TARE: transarterial radioembolization.ReferenceDesignTotal PatientsTreatments5-Year OS (%)Robles et al. [150], Spain, 2004 Retrospective, multi-center36LT30Sapisochin et al. [151], Spain, 2014Retrospective, multi-center27Neoadjuvant ethanol inj./TACE/RFA and LT51Sapisochin et al. [152], International, 2016Retrospective, multi-centerEarly:15Adv.: 33Neoadjuvant ethanol inj./TACE/RFA and LTEarly-65Adv.-45Lee et al. [153], United States, 2018Retrospective, single center17Neoadjuvant TARE/TACE/RFA and LT51.9Lunsford et al. [146], United States, 2018Prospective, single center6Neoadjuvant chemotherapy and LT83.3Wong et al. [154], United States, 2019Prospective, single5Neoadjuvant chemotherapy and TACE and LT80Krasnodebski et al. [155], Poland, 2020 Retrospective, multi-center8LT25Hue et al. [156], United States, 2021Retrospective, multi-center74Neoadjuvant chemo-radiation and LT33 KMAbdelrahim et al. [16]. United States, 2022. Retrospective, single center10Neoadjuvant chemotherapy and LT75


#### 3.3.3. Peri-Hilar CCA

When resective surgery of pCCA is considered impractical, OLT can serve as a viable and effective alternative [157]. Initial attempts at OLT for patients with pCCA yielded unfavorable results, characterized by a 5-year survival rate of approximately 20%. This did not significantly diverge from the outcomes of R1 resections or non-operative management (Table 4). The primary concern identified was an elevated incidence of disease recurrence, most frequently observed in the allograft or lungs [129]. Thus, neoadjuvant regimens have been developed to lower the disease stage and preclude recurrence and to act as an interim strategy prior to OLT [158]. A study showed that neoadjuvant chemoradiotherapy, according to the strict Mayo Clinic selection protocol, in patients undergoing OLT for pCCA resulted in a lower risk of tumor recurrence but yielded a higher incidence of hepatic vascular complications [159]. Modifications to neoadjuvant chemoradiotherapy protocols, such as the exclusion of radiotherapy, could potentially result in better outcomes for unresectable pCCA patients undergoing OLT. The therapeutic regimen encompasses external beam radiation therapy administered concurrently with 5-fluorouracil, which is then followed by bile duct brachytherapy. Subsequently, patients receive maintenance capecitabine treatment until the point of OLT [138]. As indicated by a multicenter study, the 5-year overall survival rate following neoadjuvant OLT is significantly higher compared to surgical resection, with results of 54% versus 29% [160] (Table 4, Figure 5).
cancers-16-01946-t004_Table 4Table 4Summary of the studies investigating intrahepatic cholangiocarcinoma in the liver transplant setting. KM: Kaplan–Meier; LT: liver transplant; PSC: primary sclerosing cholangitis.ReferenceDesignTotal Patients Treatments5-Year OS (%)Robles et al., [150], Spain, 2004Retrospective, multi-center23LT42Hidalgo et al., [161], United Kingdom, 2008Retrospective, single center12LT64Darwish Murad et al., [138], United States, 2012Retrospective, multi-center287Neoadjuvant chemo-radiotherapy and LT53 KMEthun et al. [160], United States, 2018Retrospective, multi-center70Neoadjuvant chemo-adjuvant and LT64 KMAzad et al. [162], United States, 2020Retrospective, single centerDe novo: 148,PSC:228Neoadjuvant chemo-adjuvant and LTDe novo: 58 KM,PSC: 74 KM

#### 3.3.4. Limitation of OLT

Despite the promising results for OLT, it has been limited by many factors, including donor organ availability, the interval between neoadjuvant therapy and OLT, which may be influenced by the center accessibility and resourcefulness, and patient factors such as blood group compatibility and presence of radiation-associated fibrosis [163,164]. Furthermore, despite the endorsement of OLT implementation for hilar CCA and pCCA, the generalization of such an approach for iCCA is controversial. For instance, almost 70% of iCCA cases recur after OLT [165]. Moreover, only a few pCCA patients are eligible for OLT, namely those with underlying liver damage related to primary sclerosing cholangitis or localized tumors with substantial biliary and/or vascular involvement [24]. In addition, the extremely poor prognosis of iCCA meant that it remaind as a contraindication for OLT in most centers [166]. Nevertheless, there are some survival benefits for OLT in early-detected unresectable iCCA; in addition, OLT can be used adjunctively with other interventions, including radiofrequency ablation and transarterial chemoembolization, for selective cases [165,166]. Living donor liver transplantation (LDLT) may prove to be another option for patients with very early iCCA.
Figure 5Outcomes of intrahepatic and peri-hilar cholangiocarcinoma by treatment modalities [16,21,66,82,152,154,155,160,161,162,167,168,169,170,171,172,173].
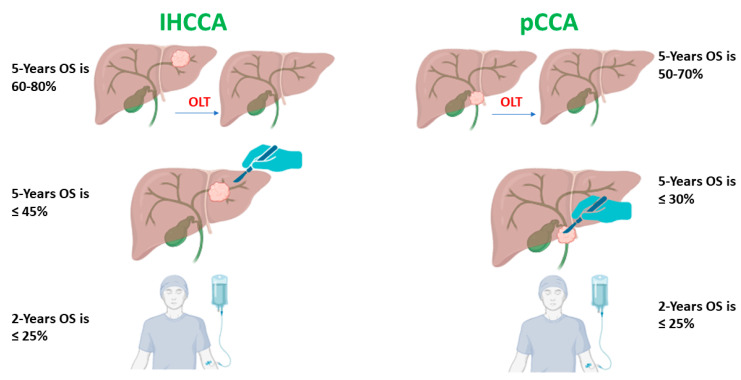


## 4. Conclusions

CCAs are aggressive primary liver tumors with an average 5-year survival of 5% in nodal-positive disease. Pre-operative multifaceted evaluations of CCAs are crucial and should include CA19-9 as a primary tumor marker, followed by cross-sectional imaging modalities such as CT and MRI to aid in surgical strategizing. Moreover, the emergence of innovative markers such as ctDNA has offered promising approaches to address the diagnostic challenges of CCA, revealing that TP53 and KRAS are among the genes with the highest frequency of mutations. Preoperative preparation is crucial, and actions such as assessing the volume of the FLR, utilizing portal vein embolization before extended hepatectomy, and optimizing the patient’s medical condition are essential for ensuring surgical feasibility and reducing the risk of complications. Additionally, staging laparoscopy is important for detecting occult metastatic disease and avoiding unnecessary surgeries. The primary surgical approach should aim for complete R0 resection along with lymphadenectomy for accurate staging. However, even in patients who have undergone tumor resection, the five-year survival rate remains low at around 25%. On the other hand, the combined neoadjuvant OLT showed an increase in the five-year survival rate when compared to surgical resection.

## Figures and Tables

**Figure 1 cancers-16-01946-f001:**
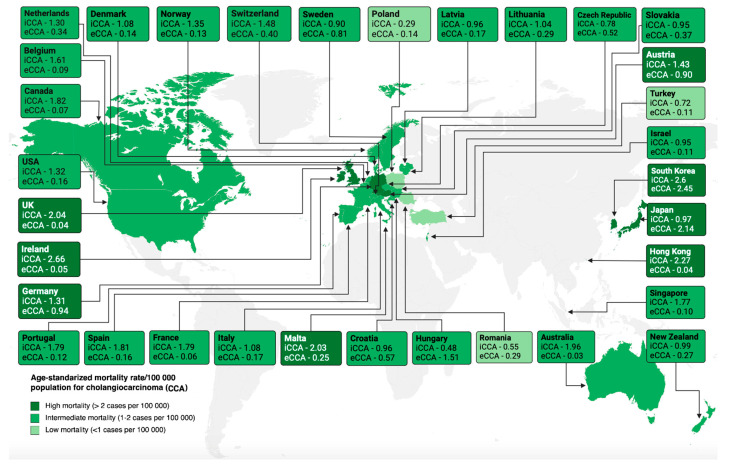
Age-standardized mortality rate/100,000 population for cholangiocarcinoma (CCA). Intrahepatic cholangiocarcinoma (iCCA), extrahepatic cholangiocarcinoma (eCCA) [3].

**Figure 2 cancers-16-01946-f002:**
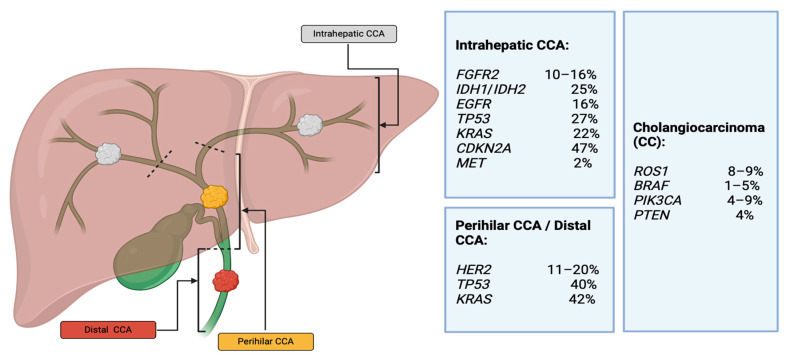
Prevalence of genetic variants in cholangiocarcinoma (CCA).

**Figure 3 cancers-16-01946-f003:**
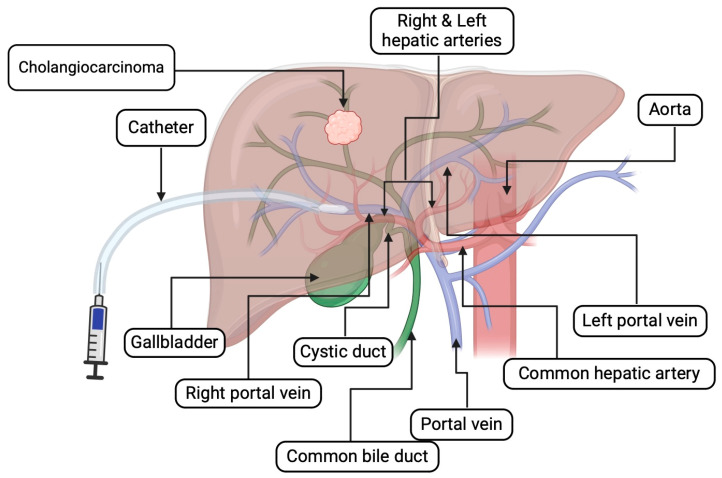
Preoperative portal vein embolization (PVE).

**Figure 4 cancers-16-01946-f004:**
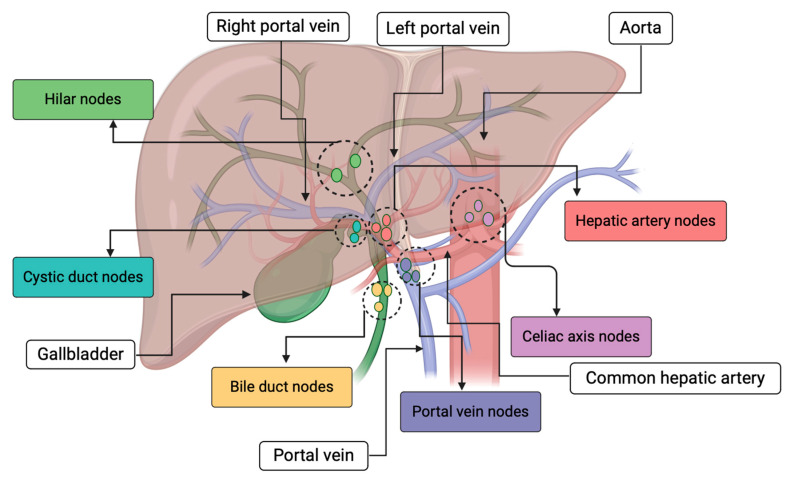
Patterns of liver lymphatic drainage. Right hemi-liver tumors drain to lymph nodes in the hepatoduodenal ligament and subsequently to peri-pancreatic and aortocaval lymph nodes. In contrast, left hemi-liver tumors drain towards lymph nodes near the left and common hepatic artery before progressing to the celiac axis.

**Table 2 cancers-16-01946-t002:** Prospective trials of orthotopic liver transplantation (OLT) in cholangiocarcinoma (CCA). Perihilar cholangiocarcinoma (pCCA), intrahepatic cholangiocarcinoma (iCCA).

Type	Characteristics		Institution/Country	Date	ID
iCCA	Early stages, cirrhotic		University Health Network, Canada	April 2018	NCT02878473
	Unresectable		Oslo University Hospital, Norway	June 2020	NCT04556214
pCCA			Oslo University Hospital, Norway	September 2021	NCT04993131
		Neoadjuvant chemo-radiotherapy	Hospital Vall d’Hebron, Spain	April 2020	NCT04378023

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
