# Peer review of "Cholangiocarcinoma: The Current Status of Surgical Options including Liver Transplantation"

_cancers, 2024, doi:10.3390/cancers16111946_

Round 1

Reviewer 1 Report

Comments and Suggestions for Authors

Esmail and colleagues wrote a review on the current status of surgical options in the treatment of cholangiocarcinoma. This is a crucial and evolving topic, however the proposed review fails to highlight the great complexity and the remaining struggles in decision making while dealing with this disease.

Major limitations are:

-       The overall presentation of data is repetitive and general, with little reference to factual evidence.

-       The role of preoperative biliary decompression is certainly a matter of debate. However, symptomatic relief is hardly an indication for it. The authors should focus more on the cholestatic impairment of liver function and the augmented risk of post-resection liver failure. On the other hand, the authors should stress on the risk associated with biliary drainage, namely bleeding, cholangitis and pancreatitis, which may interfere with the surgical program.

-       Resectability is a matter of extensive debate, and should have been addressed with more details. In particular the authors should expand on the problem of monolateral or bilateral portal or arterial involvement, as long as caval or suprahepatic involvement.

-       Surgical approach is an additional hot topic: conventional open surgery is surely the mainstay, but mini-invasive (laparoscopic and robotic) techniques are now playing an important role even for CCA. On the other hand, in selected cases, extreme procedures such as ex situ surgery cpuld be proposed if a R0 margin can be achieved. The authors should expand on this when reporting on surgical options for CCA

-       The issue of preoperative FLR optimization is very complex, especially for biliary tumors. The authors only briefly report on PVE, with no reference to other two stage procedures (ALPPS, partial ALPPS and variations, venous deprivation).

-       OLT is emerging as a promising strategy for CCA. The main focus, however, is not on leaving “a greater hepatic volume” but rather on removing the tumor with the largest possible margin, meanwhile removing the pro-carcinogenic micro-environment.

-       The role of OLT in iCCA is still under scrutiny, the authors should focus more on the ongoing debate on inclusion and exclusion criteria and on the role chemotherapy could play in this field

-       OLT for pCCA proved to offer excellent outcomes in selected patients within the Mayo protocol. This protocol, however, has several limitations that pose a great obstacle to its widespread adoption. The authors should expand on this topic.

In conclusion, this review offers a general overview on the topic of surgery for CCA, but fails to give the reader a comprehensive understanding on the complexity of the surgical management of such disease.

Author Response

Cholangiocarcinoma: The current status of surgical options including liver transplantation.

Manuscript ID: cancers-2975473

Dear Dr. Reviewer 1,

We would like to thank the learned reviewers and editor for considering our manuscript for publication and thoroughly appreciate the time taken to provide us with valuable comments to improve the readability and value of our contribution to literature. We have provided responses to all comments below.

The following issues have been addressed.

Esmail and colleagues wrote a review on the current status of surgical options in the treatment of cholangiocarcinoma. This is a crucial and evolving topic, however the proposed review fails to highlight the great complexity and the remaining struggles in decision making while dealing with this disease.

Major limitations are:

-       The overall presentation of data is repetitive and general, with little reference to factual evidence.

Response: Thank you for the feedback. We have edited it and more recent data.

-       The role of preoperative biliary decompression is certainly a matter of debate. However, symptomatic relief is hardly an indication for it. The authors should focus more on the cholestatic impairment of liver function and the augmented risk of post-resection liver failure. On the other hand, the authors should stress on the risk associated with biliary drainage, namely bleeding, cholangitis and pancreatitis, which may interfere with the surgical program.

Response: Thank you very much for this important point. We have discussed the cholestatic impairment of liver function and the augmented risk of post-resection liver failure. Also, we have stressed the risk associated with biliary drainage, namely bleeding, cholangitis, and pancreatitis.

-       Resectability is a matter of extensive debate, and should have been addressed with more details. In particular the authors should expand on the problem of monolateral or bilateral portal or arterial involvement, as long as caval or suprahepatic involvement.

Response: We appreciate your comment. We have expanded on the problem of monolateral or bilateral portal or arterial involvement, as long as caval or suprahepatic involvement.

-       Surgical approach is an additional hot topic: conventional open surgery is surely the mainstay, but mini-invasive (laparoscopic and robotic) techniques are now playing an important role even for CCA. On the other hand, in selected cases, extreme procedures such as ex situ surgery cpuld be proposed if a R0 margin can be achieved. The authors should expand on this when reporting on surgical options for CCA .

Response: We appreciate your comment. We have discussed such as ex situ surgery that could be proposed if an R0 margin can be achieved.

-       The issue of preoperative FLR optimization is very complex, especially for biliary tumors. The authors only briefly report on PVE, with no reference to other two stage procedures (ALPPS, partial ALPPS and variations, venous deprivation).

Response: Thank you for bringing this up, we have addressed this as well as have added more reports on PVE.

-       OLT is emerging as a promising strategy for CCA. The main focus, however, is not on leaving “a greater hepatic volume” but rather on removing the tumor with the largest possible margin, meanwhile removing the pro-carcinogenic micro-environment.

Response: Thank you for bringing this up, we have edited and addressed this point as the OLT is emerging as a promising strategy for CCA on removing the tumor with the largest possible margin, meanwhile removing the pro-carcinogenic micro-environment

-       The role of OLT in iCCA is still under scrutiny, the authors should focus more on the ongoing debate on inclusion and exclusion criteria and on the role chemotherapy could play in this field Response: We appreciate your comment. We have added more on the ongoing inclusion and exclusion criteria and on the role, chemotherapy could play in this field.

-       OLT for pCCA proved to offer excellent outcomes in selected patients within the Mayo protocol. This protocol, however, has several limitations that pose a great obstacle to its widespread adoption. The authors should expand on this topic.

Response: We appreciate your comment. We have discussed the limitations. As well as some discussion about the Mayo protocol OLT for pCCA that proved to offer excellent outcomes in selected patients.

Thank you!

The team

Reviewer 2 Report

Comments and Suggestions for Authors

The authors present a well-structured review of the available knowledge on various aspects of cholangiocarcinoma. It is a well-written article that may be useful for a significant number of the journal's readers, as it provides access to extensive and essential information on the subject. The conclusions are consistent with the contents presented

-The main weakness is that it does not analyze any aspect in depth, so it is of relative interest to readers already interested in cholangiocarcinoma.

-The title of the article does not seem entirely appropriate since the contents refer to a wide number of aspects that include diagnosis, staging and molecular characteristics of the tumor.

-Figure 2 offers somewhat confusing information on the location of distal cholangiocarcinoma. The indication could perfectly correspond to a cholangiocarcinoma of the middle third that could be treated by a resection of the bile duct without a pancreaticoduodenectomy, which is the treatment usually applied to distal cholangiocarcinoma.

-Figure 4 illustrating the lymph node dissemination pathways of cholangiocarcinoma appears too schematic and incomplete. It would be preferable to present the numbered lymph node groups, well described in the literature.

-The colors chosen for Figure 5 make it difficult for the reader to quickly identify each type of Bismuth's classification.

-Considering the obvious limitations on the length of the article, it is inevitable that some controversies are not well reflected in the content. However, since the article makes a specific reference to liver transplantation, I consider that it would be enriching to comment that some groups have proposed performing liver transplantation in patients with resectable hilar cholangiocarcinoma as a therapeutic option.

-The bibliographic references are adequate, but the high percentage of publications prior to 2020 contrasts with the concept of current used in the title.

Author Response

Cholangiocarcinoma: The current status of surgical options including liver transplantation.

Manuscript ID: cancers-2975473

Dear Dr. Reviewer 2,

We would like to thank the learned reviewers and editor for considering our manuscript for publication and thoroughly appreciate the time taken to provide us with valuable comments to improve the readability and value of our contribution to literature. We have provided responses to all comments below.

The following issues have been addressed.

Independent Review Report, Reviewer2

The authors present a well-structured review of the available knowledge on various aspects of cholangiocarcinoma. It is a well-written article that may be useful for a significant number of the journal's readers, as it provides access to extensive and essential information on the subject. The conclusions are consistent with the contents presented.

Response:  We appreciate your comment.

-The main weakness is that it does not analyze any aspect in depth, so it is of relative interest to readers already interested in cholangiocarcinoma.

Response:  We appreciate your comment. We have added more discussions and data as well as some figures and tables to analyze many aspects in depth.

-The title of the article does not seem entirely appropriate since the contents refer to a wide number of aspects that include diagnosis, staging and molecular characteristics of the tumor.

Response: Thank you very much for this important point. We have linked the surgical points with a wide number of aspects such as diagnosis, staging and molecular characteristics of the tumor to make this review very comprehensive and informative.

-Figure 2 offers somewhat confusing information on the location of distal cholangiocarcinoma. The indication could perfectly correspond to a cholangiocarcinoma of the middle third that could be treated by a resection of the bile duct without a pancreaticoduodenectomy, which is the treatment usually applied to distal cholangiocarcinoma.

Response: Thank you very much for this important point. We have updated and edited the figure 2.

-Figure 4 illustrating the lymph node dissemination pathways of cholangiocarcinoma appears too schematic and incomplete. It would be preferable to present the numbered lymph node groups, well described in the literature.

Response: Thank you very much for this important point. We have updated and edited figure 4 and presented the numbered lymph node groups, as well as described in the literature.

-The colors chosen for Figure 5 make it difficult for the reader to quickly identify each type of Bismuth's classification.

Response:  We appreciate your comment. We agreed and have deleted Bismuth’s figure.

-Considering the obvious limitations on the length of the article, it is inevitable that some controversies are not well reflected in the content. However, since the article makes a specific reference to liver transplantation, I consider that it would be enriching to comment that some groups have proposed performing liver transplantation in patients with resectable hilar cholangiocarcinoma as a therapeutic option.

Response:  We appreciate your comment. We have pointed out that some groups have proposed performing liver transplantation in patients with resectable hilar cholangiocarcinoma as a therapeutic option.

-The bibliographic references are adequate, but the high percentage of publications prior to 2020 contrasts with the concept of current used in the title.

Response: Thank you very much for this important point. We have added a new and recent references.

Thank you!

The team

Reviewer 3 Report

Comments and Suggestions for Authors

1. Line 52. please make sure the reference 3, Because I cannot find it for proving the Figure 1, and please tell us the reference based of Figure 1.

2. L74. It is not necessary to code 12 references to prove the outcome of LLT.

3.L94-96, the unit for CA-199 was different and please make sure and try to make it the same.

4. L129, The table 1.  Title for table will put it on the top of the table, and the format of this table need to be re-edition.

5. L148. Please tell us the means of the percentage in Figure 2. and the legend should place below the figure

6. L160. The coded reference 61 reported the results of iCCA for only CCA. Please note and add another references.

7. L192. The Figure 3 did not tell us the new information and please delet it

8. L246. Please make sure the coded reference 7 which did not mention iCCA.

9. L262-263. Please make sure to resect extra-hepatic bile duct or lymph node along the extrahepatic bile duct?

10. L320. The Bisthmus classification(Figure 5) was well known and can be deleted. If authors insist to preserve, it is necessary to code the reference.

11. L365.  it is not necessary to code 10 references from 123-133 too many to show the evidence of OLT in CCA.

12, L368, Table 2 is not necessary and suggest to provide the results briefly instead of a table.

13. L441 and 454; It is not good to repeat the survival rate.

Author Response

Cholangiocarcinoma: The current status of surgical options including liver transplantation.

Manuscript ID: cancers-2975473

Dear Dr. Reviewer 3,

We would like to thank the learned reviewers and editor for considering our manuscript for publication and thoroughly appreciate the time taken to provide us with valuable comments to improve the readability and value of our contribution to literature. We have provided responses to all comments below.

The following issues have been addressed.

Independent Review Report, Reviewer3

  1. Line 52. please make sure the reference 3, Because I cannot find it for proving Figure 1, and please tell us the reference based of Figure 1.

Response: We appreciate your valuable comment. We have added the reference to the figure.

  1. L74. It is not necessary to code 12 references to prove the outcome of LLT.

 Response: Thank you very much for this important point. We have removed some of them.

3.L94-96, the unit for CA-199 was different and please make sure and try to make it the same.

Response: We appreciate your valuable comment. We have corrected and made it the same.

  1. L129, The table 1.  Title for table will put it on the top of the table, and the format of this table need to be re-edition.

Response: We appreciate your valuable comment. We have put Title for table on the top of the table, and the format of this table has been re-edited.

  1. L148. Please tell us the means of the percentage in Figure 2. and the legend should place below the figure

Response: We appreciate your valuable comment. We have placed the legend below the figure, the percentage is for the incidence among CCA patients. 

  1. L160. The coded reference 61 reported the results of iCCA for only CCA. Please note and add another reference.

Response: We appreciate your valuable comment. We have added another reference.

  1. L192. Figure 3 did not tell us the new information and please delete it

Response: Thank you very much for this important point. We have redesigned and added more information to the figure. 

  1. L246. Please make sure the coded reference 7 which did not mention iCCA. Response: We appreciate your comment. We have added a reference.
  2. L262-263. Please make sure to resect the extra-hepatic bile duct or lymph node along the extrahepatic bile duct.

Response: We appreciate your comment. We have edited it accordingly.

  1. L320. The Bisthmus classification (Figure 5) was well-known and can be deleted. If authors insist to preserve, it is necessary to code the reference.

Response: We appreciate your comment. Agreed we have deleted the figure.

  1. L365.  it is not necessary to code 10 references from 123-133 too many to show the evidence of OLT in CCA.

Response: We appreciate your comment. Agreed We have deleted some references.

12, L368, Table 2 is not necessary, and suggest providing the results briefly instead of a table.

Response: Thank you very much for this important point. These are ongoing clinical trials and no interim analysis reported yet.

  1. L441 and 454; It is not good to repeat the survival rate.

Response: We appreciate your valuable comment. We have reviewed it. 

Thank you!

The team